# Transcriptomic Response of the Liver Tissue in *Trachinotus ovatus* to Acute Heat Stress

**DOI:** 10.3390/ani13132053

**Published:** 2023-06-21

**Authors:** Qian-Qian Li, Jing Zhang, Hong-Yang Wang, Su-Fang Niu, Ren-Xie Wu, Bao-Gui Tang, Qing-Hua Wang, Zhen-Bang Liang, Yan-Shan Liang

**Affiliations:** 1College of Fisheries, Guangdong Ocean University, Zhanjiang 524088, China; lqq10142023@163.com (Q.-Q.L.); zjouzj@126.com (J.Z.); wanghongyang0612@126.com (H.-Y.W.); wurenxie@163.com (R.-X.W.); zjtbg@163.com (B.-G.T.); wqh31016877430@163.com (Q.-H.W.); liangzhenbang0403@163.com (Z.-B.L.); yanshan-liang@outlook.com (Y.-S.L.); 2Southern Marine Science and Engineering Guangdong Laboratory, Zhanjiang 524025, China

**Keywords:** *Trachinotus ovatus*, liver, heat stress, transcriptomic analysis, gene expression

## Abstract

**Simple Summary:**

In this study, the transcriptomic response of *Trachinotus ovatus* liver was observed against high-temperature stress. Through differential expression and short time-series expression miner (STEM) analyses, some high-temperature-related genes and biological pathways were screened, which were mainly related to protein balance, hypoxia adaptation, and energy metabolism. Our results suggest that protein dynamic balance and function, hypoxia adaptation, and energy metabolism transformation are crucial in response to acute high-temperature stress. These results contribute to understanding the molecular response mechanism of *T. ovatus* under acute heat stress and provide novel insights into the selection and breeding of heat-tolerant cultivars and the high-quality development of aquaculture.

**Abstract:**

*Trachinotus ovatus* is a major economically important cultured marine fish in the South China Sea. However, extreme weather and increased culture density result in uncontrollable problems, such as increases in water temperature and a decline in dissolved oxygen (DO), hindering the high-quality development of aquaculture. In this study, liver transcriptional profiles of *T. ovatus* were investigated under acute high-temperature stress (31 °C and 34 °C) and normal water temperature (27 °C) using RNA sequencing (RNA-Seq) technology. Differential expression analysis and STEM analysis showed that 1347 differentially expressed genes (DEGs) and four significant profiles (profiles 0, 3, 4, and 7) were screened, respectively. Of these DEGs, some genes involved in heat shock protein (HSPs), hypoxic adaptation, and glycolysis were up-regulated, while some genes involved in the ubiquitin-proteasome system (UPS) and fatty acid metabolism were down-regulated. Our results suggest that protein dynamic balance and function, hypoxia adaptation, and energy metabolism transformation are crucial in response to acute high-temperature stress. Our findings contribute to understanding the molecular response mechanism of *T. ovatus* under acute heat stress, which may provide some reference for studying the molecular mechanisms of other fish in response to heat stress.

## 1. Introduction

Aquaculture is one of the fastest-growing sectors of the food industry worldwide and provides about 50% of animal protein for humans [1], which is predicted to grow to 53% by 2030 [2]. In the past few years, the aquaculture industry has developed rapidly worldwide, especially in China. However, there still are problems hindering the healthy development of the aquaculture industry, such as heat stress. The earth’s climate is rapidly warming, and spells of high temperature are becoming more frequent and severe, resulting in an overall higher rate of high-temperature stress for fish [3]. High temperatures can also accelerate the propagation of pathogens and reduce the dissolved oxygen (DO) content in water, causing severe economic losses. Fish cannot regulate body temperature, which varies with water temperature [4]. Most fish are adapted to live in a temperature range, and low and high temperatures can cause various physiological and pathological reactions, such as slow-growing, developmental delay, contour deformity, disease resistance reduction, or even death [5,6]. Thus, the study on responses to temperature stress in fish helps determine the optimal temperature of survival, growth, and development and understand the mechanisms of temperature adaptation. Related research findings would be of great value for the selection and breeding of heat-tolerant cultivars, selection of aquaculture waters, water temperature regulation of indoor aquaculture, and improvement of aquaculture efficiency.

Fish response to heat stress is highly complex. Heat stress can cause adverse effects on fish behavior, such as reduced feeding in amur sturgeon (*Acipenser schrenckii*) [7] and Atlantic salmo (*Salmo salar*) [8] and disordered respiratory rhythm, irregular swimming behavior in common carp (*Cyprinus carpio communis*) [9]. Heat stress can alter the physiological and biochemical responses of fish, including energy metabolism, oxidative stress, antioxidant, immune system, neuroendocrine, and others, which modifies biological functions. Heat stress improves the activities of glycolytic enzymes, lactase, catalase, and antioxidant enzymes in olive flounder (*Paralichthys olivaceus*) and turbot (*Scophthalmus maximus*) [10], marbled rockcod (*Notothenia rossii*) [11], gymnocypris chilianensis (*Cyprinidae*) [12], and Yangtze sturgeon (*Acipenser dabryanus*) [13], inhibits the activities of phagocytes and lysozyme in the blood of Nile tilapia (*Oreochromis niloticus*) [14] and rainbow trout (*Oncorhynchus mykiss*) [15], suppresses the secretion of sex hormones in *S. salar* [16], and exacerbates thyroid hormone disorders in Japanese medaka (*Oryzias latipes*) [17]. In addition, the content of heat shock proteins (HSPs) increases in spotted snakehead (*Channa punctata*) [18] and brook trout (*Salvelinus fontinalis*) [19] with the increase in water temperature, protecting cells from heat stress. Collectively, heat stress exerts many adverse effects on various behaviors, physiology, and biochemistry, such as increased oxidative stress, decreased immunity, and increased morbidity and mortality rates, which degrades the quality and economic benefits of cultured fish. Therefore, a scientific understanding of the effects of heat stress on fish is an inevitable requirement for the sustainable development of aquaculture industries.

Transcriptome RNA sequencing (RNA-Seq) technology has been used to understand the molecular mechanisms of fish response to heat stress, and some examples are as follows. Under heat stress, some genes related to protein folding and degradation, oxidative stress, immune response, and cell signal transduction were up-regulated in the gill tissue of chinook salmon (*Oncorhynchus tshawytscha*) [20]. Some genes associated with heat shock proteins (HSPs), fatty acid metabolism and apoptosis were also up-regulated in the liver and brain tissues of grass carp (*Ctenopharyngodon idella*) [21]. Some HSPs, ubiquitination, and immune pathways associated genes were significantly up-regulated in the liver of *A. dabryanus* [13]. These studies have shown that specific biological processes and physiological functions are activated in various fish to adapt to the high-temperature environment and reduce physiological injury. Different water areas offer varying water temperature levels. Fish have evolved specific physiological and molecular adaptations to aid survival in their respective habitats through long-term evolution and natural selection. Based on temperature tolerance levels, fish are divided into narrow temperature and eurytherm, of which eurythermal fish are divided into cold-water fish, temperature-water fish, and warm-water fish. There are some differences in the high-temperature resistance among different fish species, and conspecific fish has different responses to water temperature at different developmental stages. In general, larvae are more sensitive to changes in water temperature than adult fish. These indicate that the regulatory mechanism of heat stress response differs among different fish and the developmental stages of conspecific fish [22,23,24,25]. In summary, it is necessary to study the molecular mechanism of heat stress response in the different fish from the transcript level, and the findings are helpful for the molecular breeding of high-temperature tolerance varieties and the healthy development of the aquaculture industry.

Golden pompano (*T. ovatus*), belonging to the genus *Trachinotus*, is one of the most important species of warm-water fish and is widely distributed in coastal areas of Fujian, Guangdong, Guangxi, and Hainan provinces of China [26,27]. The optimal temperature range for *T. ovatus* is 22–28 °C [28]. The fish has become a major economically cultured marine fish along with four coastal provinces of the South China Sea due to its advantageous properties, including fast growth rate, strong disease resistance, wide environmental adaptability, and short culture cycle [29,30]. However, the frequent occurrence of extreme weather and increased culture density result in increases in water temperature, a decline in DO, rapid reproduction of pathogens, and other uncontrollable problems during the aquaculture process of this species, which in turn can result in poor physiological metabolism, a decline of stress resistance capability, increased risk of infectious diseases, etc. [31,32]. Heat stress is one of the most important environmental problems facing the aquaculture industry of *T. ovatus* and threatens the healthy and sustainable development of aquaculture [33]. Previous studies on *T. ovatus* have mainly focused on disease control [34], growth and development [35], nutrition and feed [36,37], environmental stress (salinity, hypoxia, and so on) [38,39], etc. The effect of temperature on *T. ovatus* has also been discussed, which mainly focused on larvae and juveniles [33,40,41,42,43]. These studies found that high-temperature stress improved oxygen consumption [40], reduced survival rate [41], increased jaw deformity [42], and changed the expression levels of genes related to skeletal development, nutrient digestion and absorption, protein folding, degradation, etc. [33,43]. The molecular regulatory mechanism underlying heat stress response is not yet elucidated in adults of *T. ovatus*. The liver plays a central role in fish carbohydrate, lipid, and protein metabolism and is one of the key tissues affected by high-temperature conditions [13,21,44]. Thus, it is necessary to analyze the stress response of adult *T. ovatus* liver tissue in high-temperature environments using transcriptomics and other omic technologies, which helps understand the molecular response mechanism of this species.

In this study, RNA-Seq was used to study the changes in transcription levels in the liver tissue of *T. ovatus* under acute high-temperature stress. We could screen the key genes and biological pathways closely related to high-temperature stress by differential expression and short time-series expression miner (STEM) analysis. Combined with the published research results, the main biological processes and their molecular mechanisms were clarified in response to acute heat stress in *T. ovatus*. The results of our study have importance for the selective breeding of high-temperature tolerance varieties in *T. ovatus*, the regulation of temperature and oxygen in aquaculture water, the selection of aquaculture waters, and the improvement of risk resistance in cultured fish. It can also provide a reference for the study of responding to the molecular mechanism of heat stress in other fish.

## 2. Materials and Methods

### 2.1. Heat Stress Experiment and Sample Collection

A total of 90 healthy and disease-free *T. ovatus* were purchased from Zhanjiang Hist Aquatic Technology Co., Ltd., Zhanjiang, China, and their average standard lengths (SL) and weights (SW) were 21.1 ± 1.24 cm (mean ± SD (standard deviation)) and 333.8 ± 61.21 g, respectively. After transporting to the experimental base, *T. ovatus* were evenly placed into three plastic tanks (1.0 m × 1.0 m × 1.5 m) with 30 experimental fish per tank (Figure 1). Before the formal trial, *T. ovatus* were domesticated for two weeks to minimize stress response. The domestication conditions were as follows: temperature (27.0 ± 0.5 °C), salinity (24 ± 1‰), and DO (6.0 ± 0.3 mg/L). During the domestication period, *T. ovatus* were fed twice a day with commercial pelleted feed, and 30–40% of the seawater in each tank was regularly changed every day.

Three parallel groups were set up in the experiment, and three temperature gradients, including 27 °C (control group, CT), 31 °C (high-temperature group, *HT31*), and 34 °C (high-temperature group, *HT34*), were set up in each parallel group. Before the heat stress experiment, all experimental fish were fasted for 24 h and were not fed during the experiment. After the beginning of the experiment, water temperature in three parallel groups was gradually increased from 27 °C to 34 °C at 1 °C/h and maintained at 34 °C for 2 h using heating rods (WN-B15). Three fish were selected randomly from each parallel group at 27 °C, 31 °C, and 34 °C (kept for 2 h) and anesthetized with phenolic. After measuring body length and weight, fish were dissected, and their liver tissues were removed and put into liquid nitrogen for cryopreservation. Nine liver samples were collected at each temperature (3 parallel × 3 fish/parallel = 9 fish). A total of 27 samples were collected at three temperature groups (CT, HT31, and HT34).

### 2.2. RNA Extraction, Library Construction, and Sequencing

For each temperature group, three liver samples were randomly selected for library construction and sequencing. Nine samples (3/group × 3 groups) at three temperature groups were used for all experiments and subsequent analyses. According to the manufacturer’s instructions, total RNA was extracted from nine liver samples using a Trizol reagent (Invitrogen, Carlsbad, CA, USA). Agilent 2100 Bioanalyzer (Agilent Technologies, Palo Alto, CA, USA) and RNase-free agarose gel electrophoresis were used to detect the integrity of total RNA and the presence of contamination. Qubit 2.0 Fluorometer (Invitrogen) was used to quantify RNA concentration accurately, and a NanoPhotometer spectrophotometer (Implen, Westlake Village, CA, USA) was used to detect the purity of total RNA. Only RNA samples with the required integrity, concentration, and purity were used for subsequent library construction and Illumina RNA-Seq.

The messenger RNAs (mRNAs) with poly (A) tails were enriched from qualified total RNA using magnetic beads with oligo (dT), and the enriched mRNA was fragmented into short fragments by ultrasound. The short mRNA fragments were reverse-transcribed into first-strand complementary DNA (cDNA) using random primers, and M-MuLV reverse transcriptase, followed by second-strand cDNA synthesis using RNase H and DNA polymerase I. The cDNA fragments were purified with a QiaQuick PCR extraction kit (Qiagen, Venlo, The Netherlands), end-repaired, poly (A) added, and ligated to Illumina sequencing adapters. The cDNA of about 200 bp was screened using AMPure XP beads. PCR amplification was then performed, and the PCR product was purified using AMPure XP beads to obtain a cDNA library. To obtain high-quality sequencing data, the quality of the cDNA library was strictly checked using the same method described for total RNA. The qualified cDNA library was sequenced using Illumina HiSeqTM 4000 by Gene Denovo Biotechnology Co. (Guangzhou, China).

### 2.3. Sequencing Quality Detection and Comparison

To ensure the quality of data and downstream analyses, the raw reads obtained from sequencing were filtered by fastp version 0.18.0 [45] to remove low-quality reads containing adapter sequences, reads with more than 10% of unknown nucleotide (N), all A base, and low-quality reads (more than 50% of the bases with a quality score Q ≤ 20). The quality of clean reads was evaluated by Q20, Q30, GC content, and error rate. Clean reads were mapped to the ribosome RNA (rRNA) database using the short-reads alignment tool bowtie2 (version 2.2.8) [46] to remove rRNA-mapped reads. The remaining high-quality clean reads were used for subsequent analyses.

The gene annotation file of the reference genome and gene model of *T. ovatus* was downloaded from Figshare [47], and the index of the reference genome was constructed. High-quality clean reads were mapped to the reference genome of *T. ovatus* by HISAT 2.2.4 [48], obtaining the position and annotation information on the reference genome or gene.

### 2.4. Differential Gene Expression Analysis 

The mapped reads were normalized as fragments per kilobase of transcript per million mapped reads (FPKM), and further data analyses were based on normalized data. Pearson’s correlation coefficient was calculated to evaluate the repetition correlation between parallel samples within every temperature group. Differentially expressed gene (DEG) analysis was performed using DESeq2 [49], and the genes with false discovery rate (FDR) < 0.05 and |log2 (Fold Change)| > 1 were considered significant DEGs. Gene Ontology (GO) and Kyoto Encyclopedia of Genes and Genomes (KEGG) pathway enrichment analyses were performed for all significant DEGs. The GO terms and KEGG pathways with *p* < 0.05 were considered significantly enriched.

### 2.5. STEM Analysis

Based on the transformed FPKM values, all the genes having similar expression patterns were classified by STEM [50]. The parameters used for STEM analysis were as follows: number of modules 20, correlation coefficient > 0.7, and significance threshold *p* < 0.05. GO term and KEGG pathway enrichment analyses were performed on each significant gene expression profile, understanding the specific biological processes in response to heat stress.

Based on the protein interaction data from *T. ovatus*, protein–protein interaction (PPI) network analysis of DEGs in each significant profile was performed using the String v10 [51] database to find the module network and prioritize genes. The first 300 genes were used to construct gene co-expression networks, and the network graph was visualized using Cytoscape v3.8.2 [52]. Finally, candidate hub genes were screened according to their highest connectivity with other genes.

### 2.6. Quantitative Real-Time Polymerase Chain Reaction (qRT-PCR) Validation of mRNA Expression Patterns 

For confirming the reliability of transcriptome data, 11 DEGs were randomly selected for qRT-PCR validation, and the β-actin gene was used as an internal reference gene. The gene-specific primers were designed using Primer-BLAST in National Center for Biotechnology Information (NCBI) based on the target gene sequences (Table 1). Total RNA was extracted from nine liver samples of CT, HT31, and HT34 groups using TRIzol Reagent, and RNA concentration and integrity were detected using a NanoDrop spectrophotometer (Thermo Fisher Scientific, Waltham, MA, USA) and agarose gel electrophoresis. About 100 ng RNA was used for reverse transcription using TransScript^®^ Uni All-in-One First-Strand cDNA Synthesis SuperMix for qPCR Kit (TransGen Biotech, Beijing, China) following the manufacturer’s instructions. 

The final volume of qRT-PCR reaction was 20 μL, including 8.2 μL RNase-free water, 10 μL 2× Perfect Start^®^ Green qPCR Super Mix (Beijing TransGen Biotech Co., Ltd., Beijing, China), 1 μL cDNA, 0.4 μL forward primer (10 mmol), and 0.4 μL reverse primer (10 mmol). Light Cyeler 96 (Roche, Germany) was applied for qRT-PCR, and the PCR reaction cycle program was as follows: pre-denaturation at 94 °C for 30 s, followed by 40 cycles of 94 °C for 5 s and 60 °C for 30 s. Each sample was prepared in three technical replicates, and each experiment contained a negative control without template DNA. After qRT-PCR amplification, the amplification curve and melting curve were examined to evaluate specific amplification and the relative expression levels of target genes were calculated by the 2^−△△CT^ method. The data of 11 selected genes were compared and obtained from qRT-PCR and transcriptome sequencing to validate the reliability of transcriptome sequencing data.

## 3. Results

### 3.1. Transcriptome Sequencing

A total of 386,879,166 and 58,031,874,900 bp raw data were obtained from the transcriptome sequencing of 9 samples in three temperature groups (Table 2). After removing low-quality data, a total of 386,418,164 clean reads and 57,665,720,651 bp clean data were obtained. Clean reads of CT, HT31 and HT34 groups were 44,444,572–46,596,420, 37,480,302–40,304,816 and 43,214,412–45,602,954, respectively. The clean data of CT, HT31 and HT34 groups were 6,639,750,913–6,948,044,496, 5,593,758,917–6,014,917,041 and 6,448,179,655–6,798,784,900, respectively. The Q20, Q30, GC content, and sequencing error rates of each sample were 97.81–98.11%, 93.71–94.38%, 50.12–52.84%, and 0.11–0.13%, respectively, which suggests that the sequencing quality is good. After removing reads mapped with the rRNA database, a total of 380,347,332 high-quality clean reads (44,019,362–46,521,200 from the CT group, 37,301,710–40,179,202 from the HT31 group, and 41,427,788–43,441,596 from the HT34 group) were obtained. The high-quality clean reads were aligned with the genome of *T. ovatus*, and the matching degree of nine samples was 77.72–94.03% with an average matching rate of 90.98% (Table 3), of which that of eight samples was higher than 90%, which indicates that the transcriptome sequencing data have a good alignment result with the genome. The results of sequencing, quality control, and genome comparison showed that the transcriptome sequencing data were reliable, which could ensure the accuracy of subsequent analysis.

### 3.2. Differential Gene Expression Analysis

As shown in Figure 2A, the Pearson correlation coefficients between parallel samples within the CT, HT31, and HT34 groups were 0.98–0.99, 0.93–1.00, and 0.77–0.96, respectively, indicating good reproducibility between biological replicates. Based on FPKM values, DEGs between temperature groups were screened. A total of 297 (188 up-regulated and 109 down-regulated), 842 (413 up-regulated and 429 down-regulated), and 665 (334 up-regulated and 331 down-regulated) DEGs were detected in CT vs. HT31, CT vs. HT34, and HT31 vs. HT34 comparison groups, respectively (Figure 2B). DEGs increased significantly (from 297 to 842) with the increase of water temperature (from 31 °C to 34 °C), indicating that the increasing temperature could induce changes in the expression levels of more related genes. A total of 1347 DEGs were identified in three comparison groups by temperature (Figure 2C), of which nine were shared by three comparison groups; 439 were shared by two comparison groups, and the other 899 were unique for the single comparison group. A clustering heat map of DEGs (Figure 2D) showed that 1347 DEGs had different expression patterns at three temperature points.

A total of 55 level 2 GO terms were obtained by GO enrichment analysis of 1347 DEGs, of which 24 belonged to biological processes (BP), 12 belonged to molecular functions (MF), and 19 belonged to cellular components (CC). Separate GO enrichment analysis of each comparison group (Figure 3) showed that CT vs. HT31, CT vs. HT34, and HT31 vs. HT34 had 52, 54, and 54 level 2 GO terms, respectively, of which 51 were shared by three comparison groups, and only four (cell killing of biological processes, structural molecule activity, electron carrier activity, and translation regulator activity in molecular function) were different. These results indicated that the physiological and biochemical processes related to high temperature in the liver of *T. ovatus* were activated when water temperature gradually rose to 31 °C. Although the three comparison groups had similar GO terms, the DEGs of the main GO terms increased significantly with the increase in water temperature. For example, cellular processes in biological processes contained 124 and 410 DEGs in CT vs. HT31 and CT vs. HT34, respectively. The above indicated that more genes participated in the biological processes activated at 31 °C along with a further increase in water temperature (from 31 to 34 °C).

Level 3 GO enrichment analysis showed (Appendix A) that a total of 2395, 3454, and 3109 GO terms were detected in CT vs. HT31, CT vs. HT34, and HT31 vs. HT34, respectively, of which 395, 495, and 343 were significant (*p* < 0.05). The significantly enriched GO terms included angiogenesis, antigen processing and presentation of peptide antigen via MHC class I and apoptotic process, carbohydrate metabolic process, cellular response to stress, immune system process, misfolded or incompletely synthesized protein catabolic process, regulation of protein folding, response to oxidative stress, tRNA metabolic process, proteasome-mediated ubiquitin-dependent protein catabolic process, regulation of ubiquitin-protein transferase activity, and so on.

The big categories of KEGG pathway enrichment analysis of DEGs (Figure 4A–C) showed that the significantly enriched pathways (*p* < 0.05) in three comparison groups were classified into six major categories. Metabolism, genetic information processing, and organismal systems were common in three comparison groups. Cellular processes and human diseases were common to CT vs. HT31, and HT31 vs. HT34, and environmental information processing was unique to CT vs. HT31. Environmental information processing processes, such as signaling, were primarily activated when water temperature rose from 27 to 31 °C, further activating more physiological processes in the liver of *T. ovatus*.

In the subclass KEGG enrichment analysis (Figure 4D–F), 231, 302, and 291 pathways were enriched in CT vs. HT31, CT vs. HT34, and HT31 vs. HT34, respectively, of which 36, 40, and 19 pathways were enriched significantly (*p* < 0.05). The significantly enriched pathways included biosynthesis of amino acids, one carbon pool by folate, protein processing in the endoplasmic reticulum, glycolysis/gluconeogenesis, carbon metabolism, IL-17 signaling pathway, *HIF-1* signaling pathway, proteasome, peroxisome, protein export, and so on. These significantly enriched pathways were related to carbohydrate metabolism, amino acid metabolism, environmental adaptation, immune system, translation, transcription, folding, sorting, and degradation.

### 3.3. Expression Trend and Enrichment Analysis of DEGs

A total of 1347 DEGs were clustered into eight profiles (Figure 5A), of which four profiles (4, 3, 0, and 7) were significant (*p* = 4.5 × 10^−34^, 4.0 × 10^−31^, 1.4 × 10^−10^, and 2.6 × 10^−8^). Profiles 4, 3, 0, and 7 had 315, 342, 180, and 170 genes, respectively, which were significantly higher than those of the other four non-significant profiles (49–108 genes). Four significant expression profiles showed different expression trends (Figure 5B). The expression levels of most DEGs in profile 3 and profile 4 remained unchanged as the water temperature was raised from 27 °C to 31 °C. When the water temperature reached 34 °C, the gene expression decreased in Profile 3 but increased in Profile 4. Different from Profile 3 and Profile 4, the DEGs in Profile 0 and Profile 7 were down-regulated and up-regulated at two high-temperature points, respectively. Overall, Profile 4 and Profile 7 showed up-regulation, and Profile 3 and Profile 0 showed down-regulation with the increase in water temperature.

GO and KEGG enrichment analyses were performed for DEGs in profiles 4, 3, 0, and 7 to screen key biological processes related to high temperature. The GO enrichment analysis results showed (Figure 6) that profiles 4, 3, 0, and 7 were enriched to 51, 51, 47, and 46 levels 2 GO terms, respectively. Four profiles had similar GO terms. The first four terms in biological processes were cellular processes, single-organism processes, metabolic processes pathway, and biological regulation; the first two terms in molecular function were binding and catalytic activity; the first three terms in cellular component were cell, cell part, and organelle.

GO enrichment analysis at level 3 (Appendix A) showed that DEGs in profiles 4, 3, 0, and 7 were enriched to 2453, 2171, 1760, and 1771 GO terms, respectively, in which 358, 277, 230, and 231 were significant (*p* < 0.05). The significantly enriched GO terms of DEGs in profile four mainly included carbohydrate catabolic process, angiogenesis, cellular carbohydrate metabolic process, single-organism carbohydrate metabolic process, regulation of carbohydrate metabolic process, regulation of cellular carbohydrate metabolic process, and so on. The significantly enriched GO terms for DEGs in profile 3 included antigen processing and presentation of exogenous peptide antigen via MHC class I, positive regulation of ubiquitin-protein transferase activity, positive regulation of protein ubiquitination, proteasome complex, proteasome-mediated ubiquitin-dependent protein catabolic process, and so on. The predominant enriched GO terms of DEGs in profile 0 included RNA binding, arginine N-methyltransferase activity, rRNA methyltransferase activity, tRNA methyltransferase activity, ubiquitin-like protein transferase activity, ubiquitin-protein transferase activity, and so on. The significantly enriched GO terms for DEGs in Profile 7 mainly include sperm–egg recognition, protein folding, ‘de novo’ protein folding, small molecule binding, regulation of ATPase activity, complex protein assembly, etc.

In KEGG pathway enrichment analysis (Figure 7), profiles 4, 3, 0, and 7 were enriched in 218, 237, 132, and 182 pathways, respectively, among which 27, 19, 6, and 33 pathways were significantly enriched (*p* < 0.05). The significantly enriched KEGG pathways in profile 4 included primarily amino acid metabolism, cancer, signal transduction, carbohydrate metabolism, and other related pathways, which were refined to the biosynthesis of amino acids, phenylalanine metabolism, microRNAs (miRNAs) in cancer, *HIF-1* signaling pathway, glycolysis/gluconeogenesis, and so on. The significantly enriched KEGG pathways in profile 3 included folding, sorting and degradation, translation, carbohydrate metabolism, and other related pathways, and more specific pathways included proteasome, protein export, ribosome biogenesis in eukaryotes, and so on. The significantly enriched KEGG pathways in profile 0 included primarily translation, transcription, folding, sorting and degradation, and other related pathways, which were refined to ribosome biogenesis in eukaryotes, spliceosome, ubiquitin-mediated proteolysis, and so on. The significantly enriched KEGG pathway in profile 7 included carbohydrate metabolism, amino acid metabolism, immune system, folding, sorting and degradation, and other related pathways, which could be refined to more specific pathways, such as glycolysis/gluconeogenesis, citrate cycle (TCA cycle), cysteine and methionine metabolism, antigen processing and presentation, protein processing in the endoplasmic reticulum, and so on.

The hub genes were screened from four significant profiles (4, 3, 0, and 7) by protein interaction network analysis. As shown in Figure 8, five (*PRDM10*, *MYC*, *HIF1α*, *JUNB*, and *GAPDH*), three (*PSMD8*, *PSMD3* and *SEC61AL1*), four (*DDX5*, *DKC1*, *FBL*, and *PRMT1*), and four (*HSP90AB1*, *HSPD1*, *TCP1*, and *CCT5*) hub genes were screened from profiles 4, 3, 0, and 7, respectively. The 16 hub genes had relatively high connectivity with other DEGs in the profiles, which suggests that these genes may play a relatively important role in response to acute high-temperature stress.

### 3.4. Validation of Gene Expression Patterns by qRT-PCR

To verify the DEGs identified by RNA-seq, we randomly selected 11 genes involved in HSPs (*HSP90AB1*), Ubiquitination-Proteasome (*PSMD1*, *PSMC5*, and *PSMB3*), and *HIF1A* signaling pathway (*HIF1α*, *PFKL*, *GAPDH*, *ALDOCB*, *ENO1*, *EPO*, and *MKNK2*) for qPCR. As shown in Figure 9, the relative expression levels of 11 genes were consistent with the RNA-Seq results, illustrating the accuracy and validity of our RNA-seq data and subsequent analyses.

## 4. Discussion

Based on RNA-Seq technology, 1,347 DEGs and four significant profiles were screened from the transcriptomic data of 9 liver samples of *T. ovatus*. The enrichment analysis of both DEGs and significant profiles showed that *T. ovatus* liver cells initiated biological processes, such as protein and amino acid metabolism, glucose metabolism, transcription, translation, folding, sorting and degradation to cope with acute high-temperature stress and reduce liver damage. Among these processes (Figure 10), transcriptional changes of genes related to heat shock proteins (HSPs), the ubiquitin-proteasome system (ubiquitin-mediated proteolysis and proteasome), hypoxia adaptation (*HIF-1* signaling pathway), energy metabolism (glycolysis/gluconeogenesis and fatty acid biosynthesis), etc. play very important roles in high-temperature adaptation.

### 4.1. Protein Homeostasis

#### 4.1.1. HSPs and Correct Folding of Proteins

Acute high-temperature stress can cause a cellular stress response in fish, leading to misfolding of intracellular proteins. If misfolded proteins accumulate in the body, it harms cells and interferes with their regular physiological functions [53]. HSPs are a highly conserved and ubiquitous protein family whose members participated in many important biological processes under abiotic (e.g., high-temperature, hypoxia, and heavy metal ions) and biotic stress, such as proper protein folding and processing, classical antigen presentation and cross-presentation [54,55]. By coordinating various functions of cells, HSPs protect the body from harm caused by high-temperature stress [56].

In this study, under acute heat stress, the expression of *HSP70* and *HSP90AB1* in the *T. ovatus* liver significantly increased, and the increase was positively correlated with temperature. *HSP70* is the most important member of the HSP family, with a molecular weight of around 70 KD. It is not expressed or expressed only in small amounts in cells under normal conditions, while its expression increases rapidly under stress conditions [57]. This protein is involved in folding nascent polypeptides, repairing misfolded proteins, and enhancing the heat tolerance of cells or the body, and it can speed up the restoration of protein synthesis [58]. HSP90 is one of the molecular chaperones found in cells, and its family members include HSP90α, HSP90β, and HSP83, etc. Of these, HSP90β is encoded by the *HSP90AB1* gene and participates in the regulation of cell signal transduction, protein folding, and long-term cell adaptability [59]. The upregulated expression levels of *HSP70* and *HSP90AB1* in the liver of *T. ovatus* are helpful for the refolding of misfolded proteins in cells, maintaining protein homeostasis and regular physiological functions, and minimizing damage to the fish body during heat stress. Consistent with this result, as the water temperature increased, the expression levels of *HSP70* and *HSP90* were also enhanced in brown trout (*Salmo trutta fario*) skin tissue [60], grass carp (*Ctenopharyngodon idellus*) liver tissue [61], spotted sea bass (*Lateolabrax maculatus*) liver and muscle tissue [62], *O. mykiss* head kidney and brain tissue [63,64], starry flounder (*Platichthys stellatus*) kidney tissue [65], and lake whitefish (*Coregonus clupeaformis*) liver tissue [66]. This regulation helps maintain intracellular protein homeostasis, minimizing the damage caused by high-temperature stress to the body. In summary, under high-temperature stress, fish can regulate the homeostasis of proteins by regulating the expression level of HSPs to maintain the regular physiological functions of the body and adapt to the high-temperature environment.

#### 4.1.2. Ubiquitin-Proteasome System (UPS) and Protein Degradation

In addition to HSPs, another important means to regulate protein homeostasis and function in eukaryotic cells is UPS, and the degradation of more than 80% of proteins in cells depends on this system [67]. Usually, the UPS can recognize and degrade unstable, denatured, or misfolded proteins to ensure that proteins in cells are in a homeostatic state, which is important to maintain homeostasis in cells [68]. The degradation of proteins by the UPS is an energy-consuming process consisting of a stepwise enzymatic cascade reaction. During this process, through the sequential actions of ubiquitin-activating enzyme (E1), ubiquitin-conjugating enzyme (E2), and ubiquitin ligase (E3), ubiquitin (Ub) is bound to the specific target protein, and the ubiquitinated target protein is recognized and degraded by the 26S proteasome.

The recognition and degradation of many proteins in cells depend on ubiquitination. This study found four significantly downregulated ubiquitination-related genes in *T. ovatus* liver, namely, ubiquitin-conjugating enzyme E2D4 (*UBE2D4*), ubiquitin-protein ligase E3B (*UBE3B*), photomorphogenic regulatory factor 1 (*COP1*), and ring finger and CHY zinc finger domain containing 1 (*RCHY1*) after acute heat stress treatment. The *UBE2D4* gene belongs to the ubiquitin-conjugating enzyme E2 gene family and encodes the ubiquitin-conjugating enzyme (E2), which acts as an intermediate to deliver activated ubiquitin molecules to E3 connected to the target protein [69,70]. This intermediate step is essential in the whole ubiquitination process. The *UBE3B*, *COP1*, and *RCHY1* genes encode ubiquitin ligase E3, a key enzyme in the ubiquitination reaction that binds the correct E2 and the substrate to increase the rate of ubiquitin transfer [71,72]. The gene encoding E2 in the liver tissue of antarctic eelpout (*Pachycara brachycephalum*) was suppressed at high temperatures [73], consistent with the results of this study. Under CO_2_ and high-temperature stress, the gill tissues of blue-green damselfish (*Chromis viridis*) [74] and *O. mykiss* [75] showed an enhanced ubiquitination ability by upregulating the gene encoding E2, accelerating the degradation of damaged proteins in vivo. In addition, the degradation of many proteins (e.g., eukaryotic low-density lipoprotein receptor (LDLR) and mouse p53) was inhibited, along with the reduction of E2 and E3 activities [70,76]. E2 and E3 play important roles in the degradation process of eukaryotic proteins. The downregulation of E2 and E3-related genes in the liver of *T. ovatus* suggests that the activities of E2 and E3 may be reduced due to acute high-temperature stress, which slows down the ubiquitination process of target proteins (denatured or misfolded ones).

Different Ub molecules specifically mark the target protein, and the target protein marked by Ub will be degraded into small peptides or amino acids by the 26S proteasome [68]. The 26S proteasome is a highly conserved and ATP-dependent complex enzyme composed of 19S Regulatory Particle (RP) and a 20S Core Particle (CP) [77]. The ubiquitinated protein first binds to 19S RP and is degraded in 20S CP to regulate cellular protein levels. 19S RP can be separated biochemically into the lid and base subcomplexes, among which the lid is composed of nine protein subunits, and its main function is to cut off the ubiquitin chain from the protein substrate before the protein substrate is degraded for recycling (de-ubiquitination) [78]. The Base is composed of 10 protein subunits, and its main function is to use the energy provided by ATP hydrolysis to unfold the protein substrate that has separated from the ubiquitin chain and transport it to 20S CP for degradation [79]. In this study, after high-temperature stress, the expression levels of proteasome 26S subunit non-ATPase 3 (*PSMD3*), non-ATPase 8 (*PSMD8*), non-ATPase 1 (*PSMD1*), ATPase 4 (*PSMC4*), and ATPase 5 (*PSMC5*) in *T. ovatus* liver were significantly downregulated at 34 °C. *PSMD3* and *PSMD8* encode two subunits in the lid; *PSMD1*, *PSMC4*, and *PSMC5* encode three subunits in the base. Some studies have indicated that the decreased expression of proteasome subunits leads to the loss of proteasome activity [80]. Therefore, in *T. ovatus*, downregulation of *PSMD3*, *PSMD8*, *PSMD1*, *PSMC4,* and *PSMC5* may reduce the de-ubiquitination and protein unfolding abilities of 19S RP and its protein transfer function of presenting deubiquitinated and unfold proteins to 20SCP, slowing down the subsequent protein degradation process. 20S CP is the catalytically active part of the 26S proteasome and is responsible for recognizing, stretching, and degrading deubiquitinated proteins [81]. It mainly comprises two α-rings and two β-rings, and each ring contains seven structurally similar subunits. In this study, after high-temperature stress treatment, the expression levels of proteasome 20S subunit alpha 4 (*PSMA4*), alpha 7 (*PSMA7*), beta 3 (*PSMB3*), and beta 7 (*PSMB7*) in *T. ovatus* were significantly downregulated at 34 °C, which may cause the reduction or loss of 20S CP catalytic activity. That is, the degradation of denatured or misfolding proteins was likely to slow down. In *C. idellus* intestinal mucosa and *O. mykiss* gill tissue under the conditions of oxidized fish oil and high CO_2_ concentration, respectively, *PSMA4*, *PSMB3*, and *PSMB7* were significantly upregulated, increasing the activity of 26S proteasome and helping to clear the damaged or incorrect proteins to protect tissues and cells from impairment [75,82]. The expression levels of proteasome subunit-encoding genes were increased in the muscles of rats with long-term high-intensity exercise and *C. idellus* infected with saprolegniasis, resulting in enhanced activity of 26S proteasome [83,84]. Genes encoding proteasome subunits in *C. idellus* head kidney tissue responded to high-density and saline-alkali stress mainly in negative regulation [85], similar to the results found in this study. The above results indicate that under acute heat stress, the significant downregulation of proteasome-related genes in *T. ovatus* may lead to lower levels of proteasome subunits, leading to the reduction or loss of 26S proteasome activity.

In summary, under acute high-temperature stress, the ubiquitination-proteasome system in the liver tissue of *T. ovatus* is inhibited to a certain extent. The degradation rate of misfolded proteins is reduced, allowing sufficient time for the refolding of misfolded proteins and the quality control of candidate degrading proteins. Under high-temperature stress, the concentration of DO in water decreases, and energy metabolism undergoes compensatory changes (see Section 4.2 below), yet protein synthesis and degradation consume large amounts of energy [86]. Therefore, the UPS is inhibited, representing an energy consumption reduction strategy [73]. As an important protein modification pathway, the UPS participates in multiple biological processes. However, there are few studies on UPS in fish, and the regulatory mechanisms of the genes are intricate. Thus, further in-depth exploration is required.

### 4.2. Adaptation to Hypoxia and Transition of Energy Metabolism

Water temperature is a crucial factor affecting aquatic organisms’ metabolic rate and water’s DO content. With the increase in water temperature, the metabolism of aquatic organisms is enhanced, and the oxygen consumption rate is also significantly increased. However, the temperature increase may reduce hemoglobin’s binding ability to oxygen [87]. In addition, the amount of DO in water is correlated with water temperature, i.e., an increase in water temperature leads to a decrease in DO in water, aggravating the hypoxia of aquatic organisms [88]. To adapt to the low-oxygen environment under high temperatures, some physiological activities in the metabolism of aquatic organisms may undergo adaptive changes to maintain homeostasis and normal physiological functions. In this study, with the increase in water temperature, the expression levels of genes related to hypoxia response (hypoxia-inducible factor 1, *HIF-1*), oxygen transport (oxygen transportation erythropoietin, *EPO*), and energy metabolism (phosphofructokinase L, *PFKL*; fructose-bisphosphate aldolase B, *ALDOB*; glyceraldehyde phosphate dehydrogenase, *GAPDH*; and enolase-1, *ENO1*) in the liver of *T. ovatus* showed significant changes. These changes may alter the body’s oxygen-carrying capacity and energy metabolism, resulting in adaptation to the hypoxic environment generated by acute heat stress.

*HIF-1* is a key molecule regulating the hypoxic response of tissue cells and is composed of *HIF-1α* and *HIF-1β* subunits [89]. Of these, *HIF-1α* is the active subunit of *HIF-1*. This subunit is widely present in most tissues, which are regulated by hypoxic signals and can regulate the activity of *HIF-1* [90]. The effect of *HIF-1α* depends on the oxygen concentration in the cell. Under normoxic conditions, *HIF-1α* is rapidly degraded through UPS, but under hypoxic conditions, *HIF-1α* will continue to accumulate and bind to *HIF-1β* inside the nucleus, forming active *HIF-1* [91]. This study found that the expression of *HIF-1α* in the liver of *T. ovatus* decreased at 31 °C but was significantly upregulated at 34 °C. Most tissues of channel catfish (*Ictalurus punctatus*) [92] and the embryonic tissue of zebrafish (*Danio rerio*) [93] showed a similar expression trend of *HIF-1α* (decreased first and then increased) during hypoxic stress. During the early stage of hypoxia, the expression of *HIF-1α* was not upregulated but downregulated, which may be because *HIF-1* gradually accumulated during the early stage, and this accumulation then negatively regulated gene transcription [93]. Other research indicates that hypoxic shock may also cause downregulated expression of *HIF-1α* [94,95]. Under hypoxic stress, the expression of *HIF-1α* was significantly upregulated in *D. rerio* embryos [94], oscar (*Astronotus ocellatus*) livers [96], yellow catfish (*Pelteobagrus fulvidraco*) livers [97], *O. mykiss* kidney [98], etc. It can be seen that the continuous hypoxic environment under high-temperature stress can significantly increase the expression of *HIF-1α* in *T. ovatus*. *HIF-1α* further forms different signaling pathways with various upstream and downstream proteins, mediating hypoxic signals and regulating a series of compensatory responses to hypoxia in cells.

*HIF-1α* is an important factor in regulating *EPO* expression [99]. This study showed that the expression of *EPO*, a downstream gene of *HIF-1α*, was slightly downregulated at 31 °C but significantly upregulated at 34 °C, which was consistent with the expression trend of *HIF-1α*. *EPO* is a glycoprotein hormone that can stimulate the differentiation and proliferation of erythroid precursor cells [100]. It is a key factor in regulating erythropoiesis and ensuring tissue oxygen supply [101]. The upregulation of *EPO* may improve the differentiation of red blood cells in the liver, effectively increase the number of red blood cells and hemoglobin, and then increase the body’s O_2_-carrying capacity and utilization of oxygen to compensate for the body’s hypoxia state in a high-temperature hypoxic environment [102,103]. Therefore, under high temperatures and a hypoxic environment, the upregulation of *EPO*’s expression level helps *T. ovatus* effectively use low oxygen concentrations and maintain normal physiological processes.

In addition to the *EPO* gene, the upregulated *HIF-1α* can also regulate the transcription of downstream glycolysis-related genes [104,105], which may lead to compensatory changes in energy metabolism to adapt to the hypoxic environment caused by high temperature. This study showed that when the temperature increased from 27 °C to 34 °C, the expression of genes encoding several key enzymes of glycolysis (*PFKL*, *ALDOB*, *GAPDH*, and *ENO1*) in the liver of *T. ovatus* showed significant upregulation trends. Among them, *PFKL* (a glycolytic rate-limiting enzyme) plays a decisive role in regulating the direction and rate of the glycolytic metabolic pathway and can catalyze the reaction of fructose-6-phosphate (F6P) to generate fructose-1,6-phosphate (F1,6P). Both ALDOB and GAPDH are key enzymes of glycolysis, and *ENO1* is the rate-limiting enzyme of glycolysis. F1,6P is gradually converted into phosphoenolpyruvate (PEP) with enzymes, such as ALDOB, GAPDH, and *ENO1*, which promotes the generation of more pyruvate to participate in the tricarboxylic acid cycle or other metabolic pathways. In this study, the significant upregulation of *PFKL*, *ALDOB*, *GAPDH*, and *ENO1* expressions indicates that in a high-temperature and low-oxygen environment, anaerobic glycolysis is activated in the liver of T. ovatus, becoming one of the primary energy supply pathways for the body. Similar findings have been reported in previous fish studies. In *O. Niloticus* [106], largemouth bass (*Micropterus salmoides*) [107], and large yellow croaker (*Larimichthys crocea*) [108], the livers provide energy for the body by activating the anaerobic glycolysis pathway to adapt to the hypoxic environment.

With the enhanced glycolysis, transcriptome sequencing results showed that the fatty acid metabolism-related genes long-chain acyl-CoA synthetases 1 (*ACSL1*), long-chain acyl-CoA synthetase 3 (*ACSL3*), fatty acid desaturase 2 (*FADS2*), and elongase of very long chain fatty acid 6 (*ELOVL6*) were significantly downregulated in the liver of *T. ovatus* at 34 °C, implying that fatty acid absorption and unsaturated fatty acids synthesis may be reduced. The long-chain fatty acyl-CoA synthetases (ACSLs) family is a crucial enzyme in the synthesis and catabolism of fatty acids and can catalyze the formation of acyl-CoA from free fatty acids (an energy-consuming process) [109]. *ACSL1* and *ACSL3* are important members of the ACSLs family, and they can accelerate the synthesis and catabolism of fatty acids. *ACSL1* can transport saturated fatty acids and unsaturated fatty acids into cells and catalyze them, and this protein is highly expressed in tissues related to energy metabolism, such as fat and liver tissues [110,111]. In the liver tissue of *C. idellus*, *ACSL1* catalyzes the acylation of endogenous fatty acids, activating the β-oxidation pathway and providing energy for fish [112]. *ACSL3* is a lipid droplet-associated protein that participates in the absorption of fatty acids and helps maintain lipid homeostasis. Under an acute hyperthermic hypoxic environment, the significant downregulation of *ACSL1* and *ACSL3* in the liver of *T. ovatus* may hinder fatty acid absorption and β-oxidation, thereby reducing the liver tissue’s demand for O_2_. 

In addition, the formation of unsaturated fatty acids is an oxygen-consuming process, which uses fatty acid chains as substrates to synthesize unsaturated fatty acids through a series of alternating reactions of dehydrogenation and carbon chain elongation under the catalysis of FADS and ELOVLs [113,114]. *FADS2* and *ELOVL6*, belonging to the FADS and ELOVLs families, respectively, are rate-limiting enzymes in synthesizing unsaturated fatty acids and catalyzing fatty acid dehydrogenation and carbon chain elongation reactions respectively. In this study, the significant downregulation of expression levels of *FADS2* and *ELOVL6* indicates that fatty acid dehydrogenation and carbon chain elongation were inhibited in the liver of *T. ovatus*. That is, the body’s ability to synthesize unsaturated fatty acids was reduced. Under high-temperature stress, the expression levels of genes related to unsaturated fatty acid syntheses, such as *FADS2*, *ELOVL2*, and *ELOVL5*, were also significantly decreased in the muscle of fathead minnows (*Pimephales promelas*) [115]. It can be seen that under high-temperature stress, fatty acid metabolism in fish is inhibited to a certain extent, thus reducing the body’s consumption of O_2_ and energy to adapt to the hypoxic environment and compensatory changes in energy metabolism caused by high temperature.

In summary, as the water temperature increases, the DO in the water decreases, and the continuous hypoxic environment activates the *HIF-1* signaling pathway in *T. ovatus*, which in turn regulates the cells to produce a series of compensatory responses to hypoxia, such as enhancing O_2_-carrying ability and O_2_-utilization rate, activating the anaerobic glycolysis pathway, and inhibiting fatty acid metabolism. These responses are conducive to the body’s effectively using low-concentration oxygen to maintain normal energy metabolism and other physiological processes to adapt to hypoxia caused by a high-temperature environment.

## 5. Conclusions

Through liver transcriptomic analysis, our studies found that genes and biological processes related to HSPs, UPS, HIF-1 signaling pathway, and energy metabolism (glycolysis and fatty acid metabolism) play a critical role in response to heat stress in the liver of *T. ovatus*. Under acute heat stress, the expression of HSPs was increased, and the expression of UPS was inhibited, which helped to reduce the aggregation of denatured and misfolded proteins and further maintain normal protein and liver physiological functions. Moreover, the *HIF-1* signaling pathway was activated by a hypoxic environment caused by acute heat stress, further inducing a series of compensatory responses, such as the increase in oxygen-carrying ability and oxygen utilization, activation of the anaerobic glycolytic pathway, inhibition of fatty acid metabolism, and so on, which helps effective utilization of low concentration of oxygen at the organismal level to maintain normal energy metabolism and other physiological processes. The above findings indicated that maintaining normal protein function, energy metabolism, and other physiological functions contribute to heat stress adaption. The present study preliminary explores the molecular response mechanism of *T. ovatus* under acute heat stress. These genes, pathways, and biological processes related to high temperature screened in this study would be of great value for selecting and breeding heat-tolerant cultivars and the high-quality development of *T. ovatus* farming.

## Figures and Tables

**Figure 1 animals-13-02053-f001:**
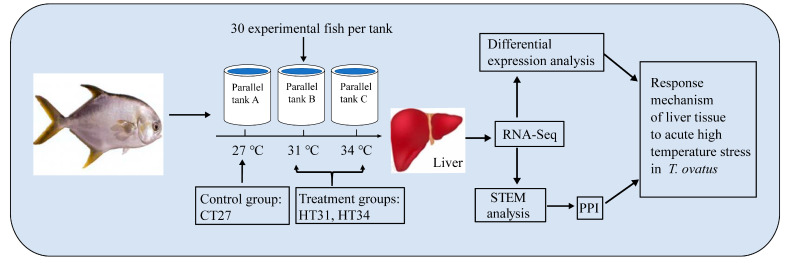
Experimental flow chart.

**Figure 2 animals-13-02053-f002:**
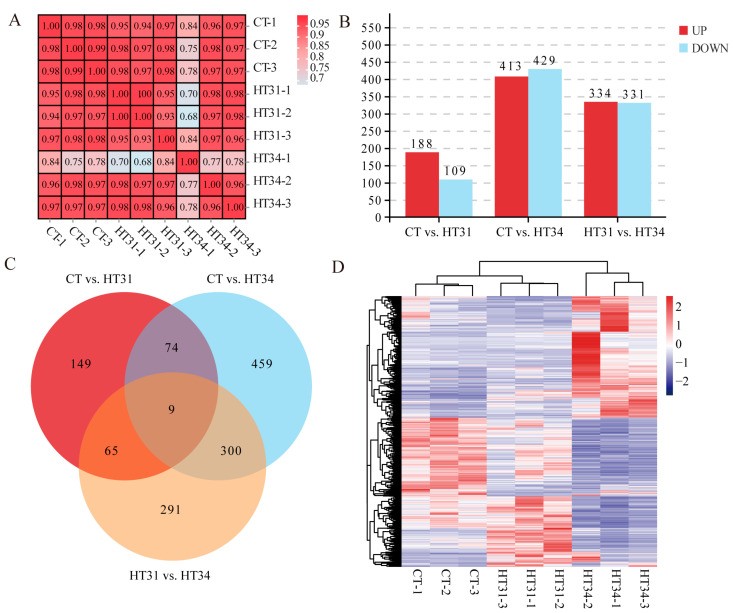
Pearson correlation coefficient heat map (**A**), numbers of up-regulated and down-regulated DEGs in three temperature comparison groups (**B**), Venn diagram of DEGs in three temperature comparison groups (**C**), and cluster heat map of 1347 DEGs (**D**).

**Figure 3 animals-13-02053-f003:**
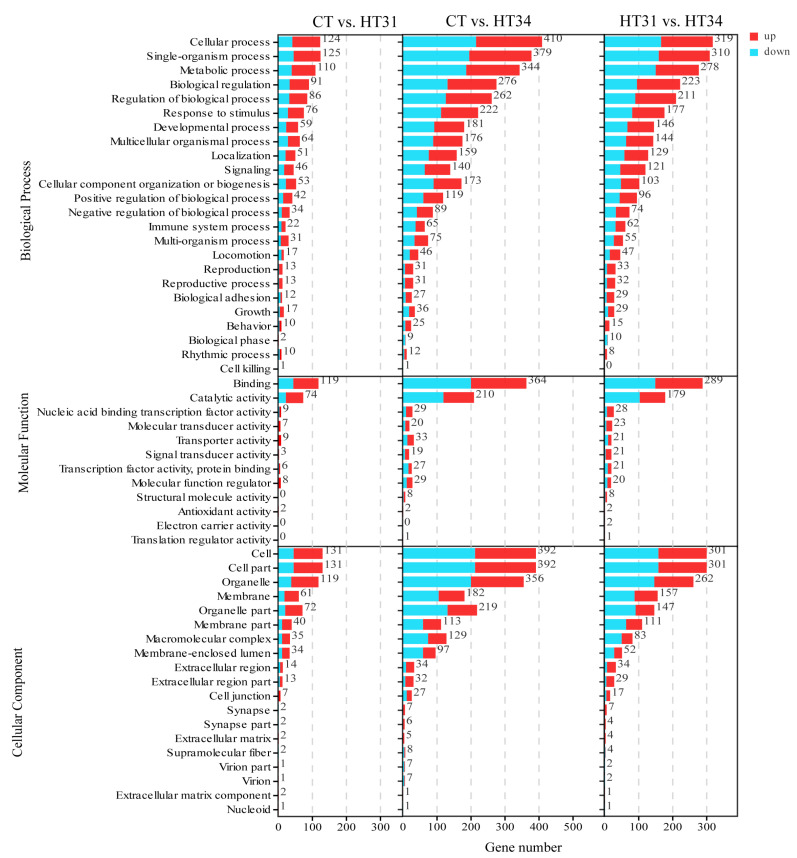
GO enrichment analysis of DEGs in CT vs. HT31, CT vs. HT34, and HT31 vs. HT34.

**Figure 4 animals-13-02053-f004:**
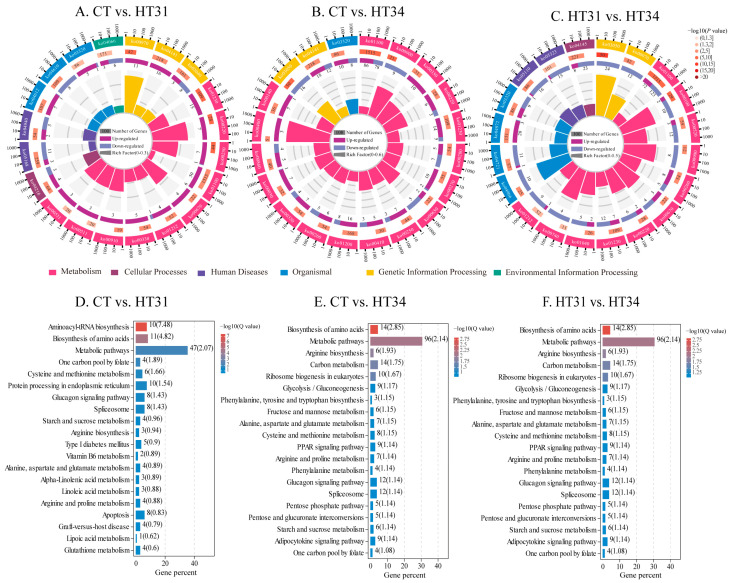
Top 20 pathways for KEGG enrichment analysis of DEGs in CT vs. HT31 (**A**,**D**), CT vs. HT34 (**B**,**E**), and HT31 vs. HT34 (**C**,**F**).

**Figure 5 animals-13-02053-f005:**
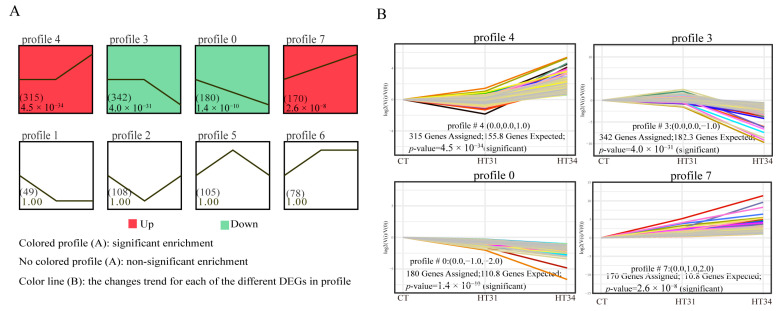
STEM analysis of all DEGs based on heat stress exposure temperature series data: (**A**) Eight expression profiles; (**B**) Expression trends for genes in profile 4, profile 3, profile 0 and profile 7.

**Figure 6 animals-13-02053-f006:**
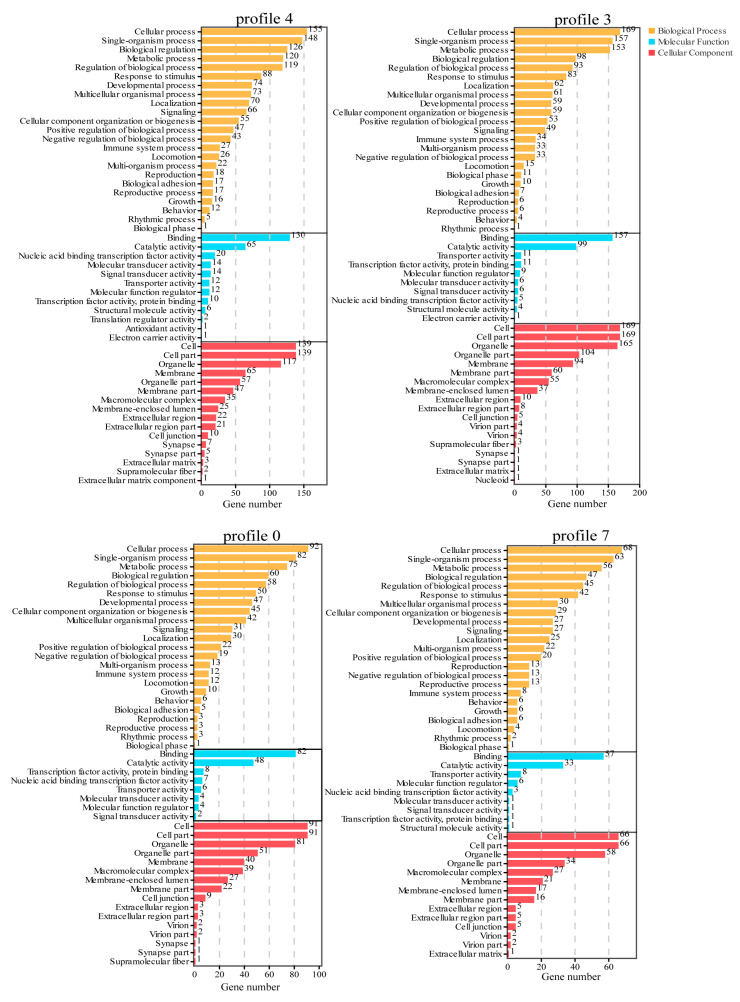
GO enrichment analysis of DEGs in Profile 4, profile 3, profile 0, and Profile 7.

**Figure 7 animals-13-02053-f007:**
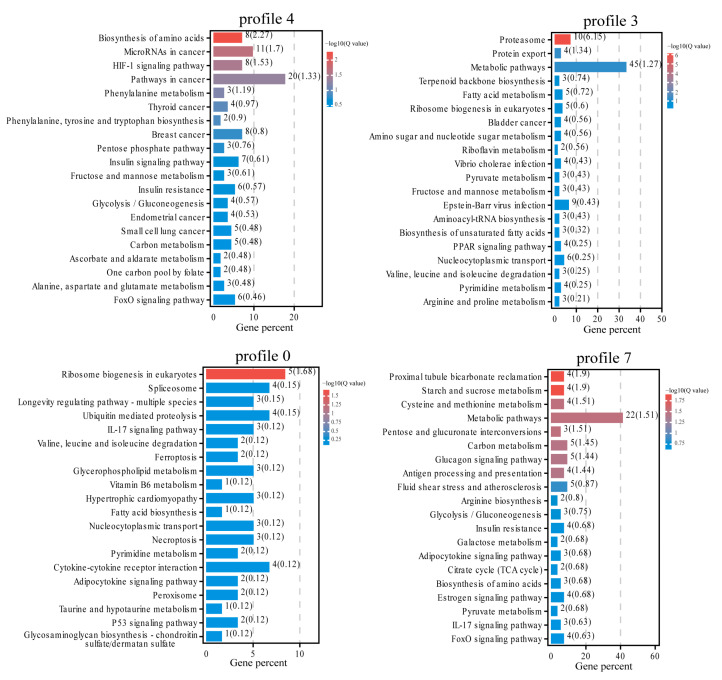
Top 20 pathways for KEGG enrichment analysis of DEGS in profile 4, profile 3, profile 0, and profile 7.

**Figure 8 animals-13-02053-f008:**
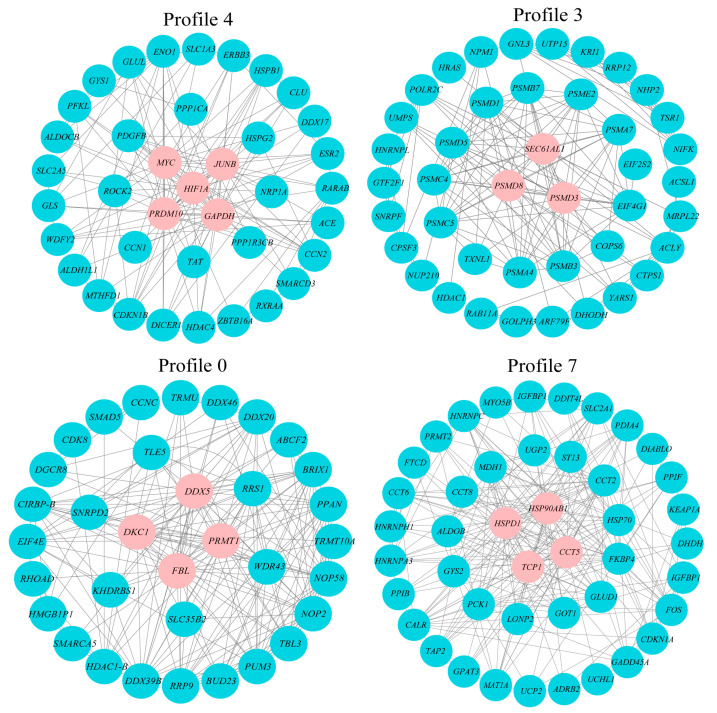
Gene co-expression networks for profile 4 (38 genes, 300 edges), profile 3 (42 genes, 300 edges), profile 0 (34 genes, 300 edges), and profile 7 (44 genes, 300 edges).

**Figure 9 animals-13-02053-f009:**
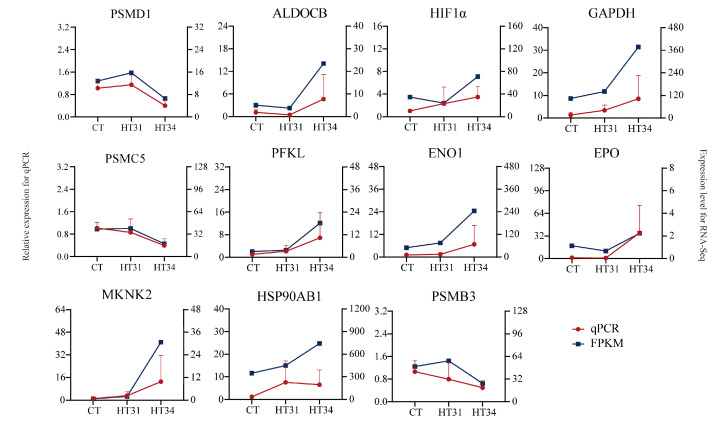
qRT-PCR analysis of eleven genes for the validation of RNA-Seq data.

**Figure 10 animals-13-02053-f010:**
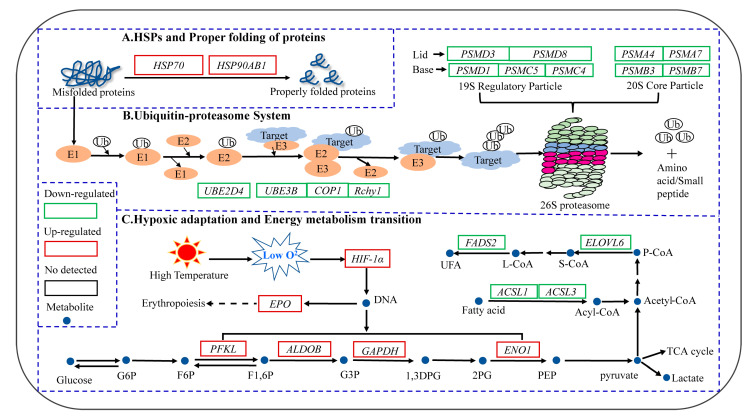
Biological process and expression changes of genes related to HSPs (**A**), the ubiquitin-proteasome system (**B**), and hypoxic adaptation and energy metabolism transition (**C**) in the liver tissue of *T. ovatus* during exposure to high-temperature stress.

**Table 1 animals-13-02053-t001:** Specific primers for eleven verified genes were used for qRT-PCR.

Primer Name	Forward Primer (5′ to 3′)	Reverse Primer (5′ to 3′)
*PFKL*	CATGTGTGCTTGGCCTCAAC	GGCCGCAGATTATACCACCA
*HSP90AB1*	CATTTCTCAGTGGAGGGCCA	GAGATGTTGAGGGGCAGGTC
*ENO1*	CTGACGACCCTAACCGCTAC	TCACCCACCACCTGGATACT
*EPO*	CTCAGCTTGCTACGGTCCTC	CACCGTCCATGTTCCCTCAA
*GAPDH*	TCGGAGTCGCAAGACAGACA	AACCTTCTTGGAGTGGAAAGCA
*ALDOCB*	TATCCCGCACTGACTCCTGA	TGTGTTCTCAACCCCGATGG
*MKNK2*	CTGTGGCTGGGAATTAGGGG	GGTTTTTGGCATCCCGAACC
*PSMB3*	TACATCGAGCCCGTGATTGC	CACACATGCCGTACATCTGC
*HIF1A*	GACACATTGGCATCACGCAG	TGTCTGCGGCTTCTTACTCG
*PSMC5*	CCCAAGGGTGTGCTGTTGTA	GCACCCTCTCCGATGAACTT
*PSMD1*	TGGCCTAGCAGTGGGAATTG	AGTGTACATTCCAGAGCGCC

**Table 2 animals-13-02053-t002:** Summary of Illumina RNA-Seq data.

Sample	Raw Reads Number	Clean Reads Number	Raw Bases (bp)	Clean Bases (bp)	Error Rate (%)	Q20 (%)	Q30 (%)	GC Content (%)
CT1	45,449,954	45,392,066	6,817,493,100	6,779,000,457	0.13	97.97	94.1	50.71
CT2	44,500,604	44,444,572	6,675,090,600	6,639,750,913	0.13	98.04	94.24	50.21
CT3	46,646,638	46,596,420	6,996,995,700	6,948,044,496	0.11	98.02	94.18	50.19
HT31-1	37,526,202	37,480,302	5,628,930,300	5,593,758,917	0.12	97.81	93.71	50.67
HT31-2	40,355,502	40,304,816	6,053,325,300	6,014,917,041	0.13	97.88	93.92	50.12
HT31-3	39,904,620	39,857,718	5,985,693,000	5,949,485,580	0.12	97.99	94.16	50.78
HT34-1	43,571,734	43,524,904	6,535,760,100	6,493,798,692	0.11	98.11	94.38	51.23
HT34-2	45,659,610	45,602,954	6,848,941,500	6,798,784,900	0.12	97.94	94	52.84
HT34-3	43,264,302	43,214,412	6,489,645,300	6,448,179,655	0.12	98.09	94.38	51.07

Notes: Error rate: the percentage of low-quality reads (based on raw Reads); Q20: the correct recognition rate of bases is 99%; Q30: the correct recognition rate of bases is 99.9%; GC content: the total number of bases G and C as a percentage of the number of four bases.

**Table 3 animals-13-02053-t003:** Results of comparison between sample and reference genome.

Sample	High Quality Clean Reads Number (%)	Unmapped (%)	Unique Mapped (%)	Multiple Mapped (%)	Total Mapped (%)
CT1	45,161,002 (99.49%)	3,409,069 (7.55%)	38,298,481 (84.80%)	3,453,452 (7.65%)	41,751,933 (92.45%)
CT2	44,019,362 (99.04%)	3,303,947 (7.51%)	37,445,636 (85.07%)	3,269,779 (7.43%)	40,715,415 (92.49%)
CT3	46,521,200 (99.84%)	2,776,762 (5.97%)	39,787,437 (85.53%)	3,957,001 (8.51%)	43,744,438 (94.03%)
HT31-1	37,301,710 (99.52%)	2,670,282 (7.16%)	31,797,678 (85.24%)	2,833,750 (7.60%)	34,631,428 (92.84%)
HT31-2	40,179,202 (99.69%)	3,327,393 (8.28%)	33,573,047 (83.56%)	3,278,762 (8.16%)	36,851,809 (91.72%)
HT31-3	39,515,610 (99.14%)	3,218,708 (8.15%)	33,369,322 (84.45%)	2,927,580 (7.41%)	36,296,902 (91.85%)
HT34-1	43,441,596 (99.81%)	2,913,743 (6.71%)	37,249,288 (85.75%)	3,278,565 (7.55%)	40,527,853 (93.29%)
HT34-2	41,427,788 (90.84%)	9,231,820 (22.28%)	29,895,198 (72.16%)	2,300,770 (5.55%)	32,195,968 (77.72%)
HT34-3	42,779,862 (98.99%)	3,236,432 (7.57%)	36,188,522 (84.59%)	3,354,908 (7.84%)	39,543,430 (92.43%)

Notes: Unmapped: the number of clean reads that unmapped onto the reference genome; Unique Mapped: the number of clean reads that uniquely mapped onto the reference genome; Multiple Mapped: the number of clean reads that mapped onto multiple loci of the reference genome; Total Mapped: the number of clean reads that mapped onto the reference genome.

## Data Availability

Illumina sequencing raw reads data and transcripts sequences have been uploaded to the NCBI SRA database; the item number is PRJNA967850.

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
