# Peer review of "Transcriptomic Response of the Liver Tissue in Trachinotus ovatus to Acute Heat Stress"

_animals, 2023, doi:10.3390/ani13132053_

Round 1
Reviewer 1 Report
This manuscript obtained the transcription library for Trachinotus ovatus liver in response to high-temperature stress and analyzed the DEGs and STEMs. It may provide novel insights into the selection and breeding of heat-tolerant cultivars and the high-quality development of aquaculture. The study was well organized and the data were informative. The manuscript is well written overall, but there are some small concerns which can make it clearer for the reader after modification.
1. In line 27 of the abstract, “liver transcriptional changes of T. ovatus were investigated under acute high-temperature stress (27℃, 31℃, and 34℃) using RNA sequencing (RNA-Seq) technology.”, the temperature 27℃ group was as control, it should be described clearly.
2. Why the author chose liver tissue, there should be some explanation in the article.
3. What is the optimal temperature for golden pompano? Please add the information in line 104.
4. Did the qRT-PCR perform triple times?
Reviewer 2 Report
General Comments
The manuscript entitled “Based on the transcriptome analysis to reveal the response 2 mechanism of liver tissue to acute heat stress in Trachinotus 3 ovatus.” is a very interesting work, with elucidative schemes and with a lot of bibliographic support. However, I have some general doubts that I would like clarification:
Why did authors choose these temperatures?
What is the effective n per group? If it's just n = 3, isn't it too small to draw indications? What were the statistical differences observed to conclude that there was an increase, inhibition, ...
Specific Comments
To improve the quality of manuscript, I have suggested the following points:
1- Line 108-115 - include references
2- Line 115 -117 - information not relevant to the subject
3- Line 117-118 - statement without reference
4- Line 118-121 - information relevant to the subject but that needs to be better detailed and clarified
5- Tables would be easier to read with legends for abbreviations.
Author Response
Dear reviewer,
Dear Reviewer,
The manuscript entitled “Based on the transcriptome analysis to reveal the response mechanism of liver tissue to acute heat stress in Trachinotus ovatus.” is a very interesting work, with elucidative schemes and with a lot of bibliographic support. However, I have some general doubts that I would like clarification:
Our answers: Thank you very much for your letter and recognition of our manuscript “Based on the transcriptome analysis to reveal the response mechanism of liver tissue to acute heat stress in Trachinotus ovatus” (animals-2422576). We will hard study your advice and make corrected modifications on the manuscript.
Why did authors choose these temperatures?
Our answers: We thank the reviewer of pointing out this issue. The optimal temperature range for golden pompano is 22 – 28℃ (Peng, 2007; Liu, 2013; Liu, 2022), and the fish has become the dominant fish in modern marine ranch farming in the South China Sea. However, the frequent occurrence of extreme weather and increased culture density of T. ovatus can cause some uncontrollable problems, including higher water temperature, lower DO, and rapid reproduction of pathogens, which will further affect the growth and development of T. ovatus. In our preliminary experiments, we found that fish died at 36℃. Therefore, we choose 27℃ as control group, 31℃ and 34℃ as experimental groups, to deeply understand the influence of temperature on the T. ovatus and provide scientific basis for the aquaculture industry in South China.
What is the effective n per group? If it's just n = 3, isn't it too small to draw indications? What were the statistical differences observed to conclude that there was an increase, inhibition, ...
Our answers: We agree that more study or more data would be useful for our study. Based on previous studies (Huang et al., 2018; Yang et al., 2021), three samples per group is effective in this study. In generally, three biological replicates were effective for the transcriptome studies of aquatic animals, such as chinook salmon (Oncorhynchus tshawytscha) (Tomalty et al., 2015), grass carp (Ctenopharyngodon idella) (Zhang et al.,2022), and yangtze sturgeon (Acipenser dabryanus) (Chen et al.,2023). In our study, sequencing results, quality control, and genome comparison showed that transcriptome sequencing data were reliable, and the Pearson correlation coefficient data showed good reproducibility within the group. Thus, our experimental data is accuracy and reliability.
The statistical differences were detected by differential gene expression analysis using DESeq2 software package (Love et al., 2014). Through observing the value of “false discovery rate (FDR)” and “Fold Change (FC)”, we can conclude that there was an increase or inhibition. For example, when the FDR < 0.05 and log2 (FC) > 1, it suggested that the expression of gene is increased and those genes were defined as up-regulated differentially expressed genes; when the FDR < 0.05 and log2 (FC) < -1, it suggested that the expression of gene is inhibited and those genes were defined as down-regulated differentially expressed genes.
Thanks very much for your constructive suggestions.
Specific Comments. To improve the quality of manuscript, I have suggested the following points:
1- Line 108-115 - include references
Our answers: Thanks very much for your reminding. As suggested by the reviewer, we have read the reference carefully and added corresponding reference in the line 110-117 of our resubmitted manuscript.
2- Line 115 -117 - information not relevant to the subject
Our answers: Thank you for your reminder. The contents of line 117 -119 in the resubmitted manuscript can clearly tell readers where the research directions of domestic and foreign scholars about the T. ovatus are mainly concentrated. Those helps readers to write further the impact of temperature on the T. ovatus, also shows that the report on the impact of acute high temperature stress on the T. ovatus is lacking, thus further underlining the importance of our study. Thanks again for your reminder.
3- Line 117-118 - statement without reference
Our answers: Thanks very much for your careful checks. The contents in line 119-120 of the resubmitted manuscript was summarized from the references 33 and 36-40. Based on your suggestion, we have added reference in our resubmitted manuscript (line 120).
4- Line 118-121 - information relevant to the subject but that needs to be better detailed and clarified
Our answers: Thanks for your suggestion. The contents of lines 119-120 and 121-124 are rather intimately connected to each other. Lines 121-124 further elaborate important information in references 33 and 40-43. We have also added references to the corresponding content in the line 121-124 of our resubmitted manuscript. Thank you very much again.
5-Tables would be easier to read with legends for abbreviations.
Our answers: Thanks for your suggestion. About the abbreviated legend, we have explained in detail below the table 2 and table 3, respectively.
Thank you very much for your professional questions and positive suggestions on the reference of the article. We checked the references again and made some corresponding modifications in our article.
References:
Peng, Z.D. Seawater quality variety-Biological characteristics and breeding of Trachinotus ovatus. Aquaculture of Beijing, 2007, 04:41-43.
Liu, R.J. Effects of salinity and temperature stress on physiological function of selective group of Trachinotus ovatus. Shanghai Ocean Univ. 2013.
Liu, Q. Effect of flow velocity on swimming behavior of Trachinotus ovatus. Guangdong Ocean Univ. 2022. doi:10.27788/d.cnki.ggdhy.2022.000336.
Huang, J.; Li, Y.; Liu, Z.; Kang, Y.; Wang, J. Transcriptomic responses to heat stress in rainbow trout Oncorhynchus mykiss head kidney. Fish Shellfish Immunol. 2018, 82, 32–40. https://doi.org/10.1016/j.fsi.2018.08.002
Yang, T.; Zhang, Y.; Meng, W.; Zhong, X.; Shan, Y.; Gao, T. Comparative transcriptomic analysis brings new insights into the response to acute temperature acclimation in burbot (Lota lota lota). Aquac. Rep. 2021, 20, 100657. https://doi.org/10.1016/j.aqrep.2021.100657
Tomalty, K.M.H.; Meek, M.H.; Stephens, M.R.; Rincón, G.; Fangue, N.A.; May, B.P.; Baerwald, M.R. Transcriptional response to acute thermal exposure in juvenile chinook salmon determined by RNAseq. G3 Genes Genom. Genet. 2015, 5, 1335–1349. doi:10.1534/g3.115.017699.
Zhang, W.; Xu, X.; Li, J.; Shen, Y. Transcriptomic analysis of the liver and brain in grass carp (Ctenopharyngodon idella) under 763 heat stress. Mar. Biotechnol. 2022, 24, 856–870. doi:10.1007/s10126-022-10148-6.
Chen, Y.Y.; Wu, X.Y.; Lai, J.S.; Liu, Y.; Song, M.J.; Li, F.Y.; Gong, Q. Integrated biochemical, transcriptomic and metabolomic 742 analyses provide insight into heat stress response in yangtze sturgeon (Acipenser dabryanus). Ecotoxicol. Environ. Saf. 2023, 743 249, 114366. doi:10.1016/j.ecoenv.2022.114366.
Love, M.I.; Huber, W.; Anders, S. Moderated estimation of fold change and dispersion for RNA-seq data with DESeq2. Genome Biol. 2014, 15, 550. doi:10.1186/s13059-014-0550-8.

Round 2
Reviewer 2 Report
I believe that the suggestions and doubts posed to the authors were satisfactorily answered.